# Use of Clopidogrel, Prasugrel, or Ticagrelor and Patient Outcome after Acute Coronary Syndrome in Austria from 2015 to 2017

**DOI:** 10.3390/jcm9113398

**Published:** 2020-10-23

**Authors:** Safoura Sheikh Rezaei, Andreas Gleiss, Berthold Reichardt, Michael Wolzt

**Affiliations:** 1Department of Clinical Pharmacology, Medical University of Vienna, Waehringer Guertel 18-20, 1090 Vienna, Austria; safoura.klopprogge@meduniwien.ac.at; 2Center for Medical Statistics, Informatics, and Intelligent Systems, Medical University of Vienna, Spitalgasse 23, 1090 Vienna, Austria; andreas.gleiss@meduniwien.ac.at; 3Austrian Health Insurance Fund, Burgenland, Siegfried Marcus-Straße 5, 7000 Eisenstadt, Austria; berthold.reichardt@oegk.at

**Keywords:** ACS, DAPT, P2Y12 inhibitors, pharmacoepidemiology

## Abstract

Background: Dual antiplatelet therapy improves patient outcome after acute coronary syndrome (ACS), but prescription differences of P2Y12 inhibitor treatments exist. The aim of the present investigation was to study the long-term utilization and patient outcomes of clopidogrel, prasugrel, and ticagrelor in patients with ACS from 2015 to 2017 in Austria. Methods: Data from 13 Austrian health insurance funds of patients with a hospital discharge diagnosis of ACS for the years 2015 to 2017 were analyzed. The primary end point was to investigate the recurrence of ACS or death. Results: Of 49,124 P2Y12 inhibitor-naive patients with a hospital discharge diagnosis of ACS, 25,147 subjects filled a P2Y12 inhibitor prescription within 30 days after the index event. Of these patients, 10,626 (42.9%) subjects had a prescription for clopidogrel, 4788 (19.3%) for prasugrel, and 9383 (37.8%) for ticagrelor. Ticagrelor was the most frequently prescribed P2Y12 inhibitor among patients below 70 years old, and clopidogrel in those aged ≥70 years. Occurrence of an endpoint was highest in elderly patients. After adjustment for age, sex, and pre-existing medication as proxy for comorbidity, the hazard ratio for ACS or death for prasugrel vs. clopidogrel of 0.70 (95% CI: 0.61; 0.79) was similar to that of ticagrelor vs. clopidogrel (0.70; 95% CI: 0.64; 0.77). Conclusion: Prescription of ticagrelor or prasugrel after ACS were associated with a lower risk of ACS recurrence or death compared to clopidogrel.

## 1. Introduction

Acute coronary syndrome (ACS) is the leading cause of morbidity and mortality in developed countries [1]. The combination of treatment with a platelet P2Y12 receptor inhibitor, such as clopidogrel, prasugrel, or ticagrelor with acetylsalicylic acid (dual antiplatelet therapy, DAPT), has been shown to prevent coronary peri-procedural thrombotic complications and recurrent ischemic events [2,3]. Randomized controlled trials have shown a reduced reoccurrence of ACS in patients treated with a DAPT comprising prasugrel or ticagrelor compared to clopidogrel [4,5].

According to current guidelines, DAPT consisting of acetylsalicylic acid and a P2Y12 inhibitor, preferably prasugrel or ticagrelor, is recommended for 12 months [6,7]. Epidemiological studies have reported a smaller number of cardiovascular (CV) events or deaths in patients with prasugrel or ticagrelor [8,9,10]. Of note, clopidogrel is more frequently used in older patients and in those who have more comorbidities [11,12]. One of the most debated questions regarding DAPT relates to its optimal prescription duration [9,13]. Previous reports have also highlighted the impact of gender on clinical outcomes in patients presenting with ACS [14,15,16]. Several studies have reported an underuse of evidence-based treatment in women with ACS, who were also more likely to have a delayed reperfusion time [15,16].

The aim of this observational study was to investigate epidemiological data on P2Y12 inhibitor utilization after an ACS in Austria between 2015 and 2017 and the association between the choice of therapy (clopidogrel, prasugrel, or ticagrelor) and recurrence of ACS or death.

## 2. Materials and Methods

This study was approved by the Ethics Committee of the Medical University of Vienna (EK-No. 2042/2018) and performed according to the Declaration of Helsinki. Patients’ informed consent was not required due to the retrospective design of the present study. The present investigation was performed in cooperation with the Pharmacoeconomics Advisory Council of the Austrian Sickness Fund.

### 2.1. Data Preparation

The public health insurance system in Austria provides health care for about 99% of the Austrian population with its different regional funds. We analyzed data between 2015 and 2017 from medical services covered by these health insurance funds. These data include patients’ demographic data, information on hospital discharge diagnoses using the International Classification of Diseases (ICD) system, and reimbursed drug prescriptions. Data were pseudonymized and data storage and handling were in agreement with Austrian privacy laws.

Adult patients (aged 18 years or older) who were discharged with the principal diagnosis of ACS (ST elevation myocardial infarction (STEMI); non-ST elevation myocardial infarction (NSTEMI)) by following ICD-10 codes I24.9, I25.0, I25.1, I20, and I.21, and subcodes, were included. Of these patients only those who filled a prescription of P2Y12 inhibitors (clopidogrel, prasugrel, or ticagrelor) within 30 days after the index ACS diagnosis were selected. Patients who had already been under P2Y12 inhibitor treatment during the period of 120 days before index diagnosis were excluded. For simplification, we excluded the small number of patients with medical records indicating simultaneous prescription of several P2Y12 inhibitors during the observational period. Subsequent hospitalizations for ACS within 30 days were interpreted as a single ACS episode.

### 2.2. Statistical Methods

Metric variables were reported by medians and interquartile ranges (IQR) due to non-normal distribution (graphically checked) and compared between P2Y12 inhibitors using a Kruskal–Wallis test. Categorical variables were described by absolute and relative frequencies and compared between P2Y12 inhibitors using Chi-square tests. The duration of clopidogrel, prasugrel, or ticagrelor intake was defined as the time between the first and last filled prescription increased by daily doses per volume for the each prescribed P2Y12 inhibitor. Based on this definition, we estimated drug survival by using product-limit method and censored data for death and end of data availability. The distribution of time to event, which was defined as re-admission for ACS or death, was estimated using the product-limit method, censoring for end of data availability, and compared between P2Y12 inhibitors using the log-rank test. The starting point for survival analyses was defined 30 days after the first prescription of clopidogrel, prasugrel, or ticagrelor after discharge from index ACS. Patients who were never at risk, i.e., for whom data availability ended or who died within 30 days after discharge from hospital or after first prescription, were excluded from this. Additionally, readmitted ACS patients within 30 days after the first prescription of P2Y12 inhibitors were excluded. To estimate crude and adjusted hazard ratios with 95% confidence intervals, we used univariate and multivariate Cox regression models, respectively. The following adjustment variables were used for the multivariate Cox regression model: sex, calendar year of index admission (as categorical variable), the interaction of age and sex, four binary covariables indicating prescription of anti-diabetic drugs (ATC codes A10A or A10B), anti-obstructive medicine for chronic obstructive lung disease (R03), ß-blockers, ACE inhibitors, or ARB as CV medication (ATC codes C07 or C09), and lipid-lowering medication defined by HMG CoA reductase inhibitors as (ATC code C10AA) within 120 days prior to the index ACS. Furthermore, we adjusted for age at treatment start, considering the non-linear associations by natural cubic spline basis with five degrees of freedom. Hazard ratios for age were derived from these models by using the quartile medians (Q1: 51 years, Q2: 61 years, Q3: 72 years, and Q4: 82 years). The median of Q1 was used as reference. As a random factor we included the origin of data, i.e., one of the 13 health insurance funds. In addition, hospitalization with ACS and death, respectively, were considered as separate endpoints in univariate and multivariate Cox regression models. In addition, for the analysis of hospitalization with ACS, death was treated as censoring event. Median follow-up time was estimated using the reverse Kaplan–Meier method [16]. In the present study, 95% confidence intervals excluding parity or two-sided *p*-values < 0.05 were considered as indicating statistical significance. All statistical analyses were performed with the Statistical Analysis System, version 9.4 (SAS Institute Inc., Cary, NC, USA).

## 3. Results

Forty-nine thousand one hundred twenty-four P2Y12 inhibitor-naive patients with ACS were identified (Figure 1). We excluded 195 patients who died or had a re-ACS within 30 days after first prescription of P2Y12 inhibitors after index ACS, since reimbursement data are capped on a quarterly basis. Additionally, in some cases patients are discharged with a promotional offer of one month’s free medication, and we therefore used the time frame of 30 days.

Of 49,124 patients, 25,147 subjects filled a prescription of a P2Y12 inhibitor within 30 days after the index event. In total, 10,626 (42.9%) subjects were detected with clopidogrel, 4788 (19.3%) with prasugrel, and 9383 (37.8%) with ticagrelor. Three hundred and fifty patients with simultaneous prescription of P2Y12 inhibitors were excluded. Of these, 250 patients switched between clopidogrel and ticagrelor, 71 patients between clopidogrel and prasugrel, 57 patients between ticagrelor and prasugrel, and 7 patients between clopidogrel, ticagrelor, and prasugrel.

Patient characteristics are summarized in Table 1. The number of patients receiving prasugrel or clopidogrel decreased from 22% to 17% and from 44% to 41% between 2015 and 2017, respectively. At the same time an increase in the number of patients taking ticagrelor was noted (Table 2). Clopidogrel was the most frequently prescribed P2Y12 inhibitor in patients ≥70 years.

### 3.1. Drug Survival of P2Y12 Inhibitor Therapy

Drug survival is shown in Table 2. During the observation period of 24 months, the median drug survival for clopidogrel, prasugrel, and ticagrelor was 11.8 (IQR 6.4–17.0), 12.1 (IQR 11.3–13.5), and 12.0 (IQR 11.1–12.9) months, respectively.

### 3.2. P2Y12 Inhibitor Therapy and Clinical Outcome

For this analysis, data from 23,816 patients with 2910 events during a median follow-up period of 23 months (IQR 14.6; 31.7) were available. Figure 2A shows event-free survival by type of therapy. 1760 patients had a recurrence of ACS and 1150 patients died without a recurrent ACS diagnosis. The cumulative incidence for ACS or death at 36 months was 23.0% in patients receiving clopidogrel, and 10.6% and 13.7% in those receiving prasugrel or ticagrelor, respectively.

Adjusting for sex, age, their interaction, pre-existing medication, and calendar year as proxy for comorbidity (Table 3), the effect of medication on the composite endpoint was different when comparing the novel P2Y12 inhibitors (prasugrel and ticagrelor) with the standard P2Y12 inhibitor (clopidogrel), with a hazard ratio (HR) of 0.70 (95% CI: 0.61; 0.79) for prasugrel vs. clopidogrel and 0.70 (95% CI: 0.64; 0.77) for ticagrelor vs. clopidogrel. A re-parameterization of the adjusted model allows clopidogrel and prasugrel to be compared with ticagrelor, and results in an HR 1.76 (95% CI 1.62; 1.91) for clopidogrel vs. ticagrelor and 0.86 (95% CI 0.76; 0.97) for prasugrel vs. ticagrelor.

In a separate statistical analysis for each component of the composite end point, the adjusted hazard ratio for recurrence of ACS for ticagrelor vs. clopidogrel was 0.76 (95% CI: 0.68; 0.85) and for prasugrel vs. clopidogrel 0.83 (95% CI: 0.72; 0.95). There was a small increase in the recurrence of ACS in 2017 compared to 2015 (crude HR 1.14; 95% CI: 1.03; 1.30). The adjusted hazard ratio for death showed a lower hazard ratio for prasugrel or ticagrelor versus clopidogrel (0.35 (95% CI: 0.26; 0.49) and 0.62 (95% CI: 0.54; 0.72), respectively).

Event-free survival by type of treatment for the subgroup of patients ≤75 years was also analyzed (Figure 2B). Of this subgroup, 1282 patients had a recurrence of ACS and 354 patients died without recurrent ACS diagnosis. The adjusted hazard ratio for recurrence of ACS or death was similar for ticagrelor vs. clopidogrel (0.73 (95% CI 0.65; 0.82)) and prasugrel vs. clopidogrel (0.73 (95% CI 0.64; 0.84)).

Endpoints were detected more frequently in women compared to men (Table 3). The interaction of anti-diabetic medicines and P2Y12 inhibitor medication was not added to the model in Table 3, since it was not statistically significant (*p* = 0.260). A further analysis investigated the non-linear continuous age effect in female and male patients. The hazard at the median age of patients in the second quartile group, which included patients between 55 and 66 years old, was compared to that of the median age of patients below 55 years old. After adjustment, male patients of 61 years of age had approximately the same hazard for event-free survival as male patients aged 51 (HR, 1.02; 95% CI, 0.88; 1.18) (Figure 3A). On the other hand, female patients with a median age of 61 years had a significantly lower hazard for event-free survival compared to female patients aged 51 years (HR, 0.63; 95% CI, 0.48; 0.84) (Figure 3B).

## 4. Discussion

This retrospective epidemiological study on drug utilization of P2Y12 inhibitors after ACS revealed two new findings. Our first major finding is the numerically lower cumulative incidence of ACS or death in patients receiving prasugrel than in those with ticagrelor or clopidogrel. Similar results were reported recently in a randomized controlled trial (ISAR REACT5) [17], which demonstrated superiority of prasugrel over ticagrelor in patients with ACS on the composite endpoint of myocardial infarction, death, or stroke. However, in the present retrospective analysis, this salutary effect of prasugrel over ticagrelor on the composite of ACS or death was not evident after correction for co-factors and did not precipitate when a limitation by age ≤75 years was applied. While the majority of events in elderly patients were deaths, the composite endpoint was driven by recurrence of ACS in patients aged ≤75 years. Several differences are obvious between the two trials, in addition to the retrospective and non-randomized design of the present analysis and the components and definition of the composite endpoint. The present study population was selected by discharge diagnosis alone. In ISAR REACT5, randomization was balanced regarding treatment allocation and assignment to prasugrel was not limited by age: in eligible elderly subjects, the daily dose of prasugrel was reduced. In contrast, only a very small proportion in our cohort received a prescription of prasugrel and an even smaller fraction of elderly patients, which introduced a bias into the crude analysis. Further, the number of trial participants who discontinued the open-label treatment with prasugrel or ticagrelor early was below 30% in ISAR REACT5, whereas approximately 49% of our patients did not fill a prescription for a P2Y12 inhibitor within 30 days after the index event.

Another real-word European study (RENAMI Registry) reported the clinical benefit of prasugrel use in patients with ACS after percutaneous coronary intervention (PCI) and discussed the superiority of prasugrel over ticagrelor in patients with NSTEMI after propensity score matching [18]. Further, a previous prospective, observational, multicenter cohort study presented lower cardiovascular adverse events for NSTEMI patients taking prasugrel compared to those with ticagrelor after undergoing a PCI [10]. The present analysis did not distinguish between STEMI and NSTEMI and was different in study design and endpoint definition compared to these previous studies.

Interestingly, comparing the present findings with our own previous study employing a similar study design, the clinical benefit of prasugrel over clopidogrel on the composite or separate end points was less pronounced. Together with the smaller percentage of patients receiving clopidogrel, this indicates a less selective prescription of prasugrel compared to the earlier clinical drug utilization pattern. Similar trends of patient selection when prescribing P2Y12 inhibitors have been reported in other observational studies [10,19]. In the present study, the prescription trend of P2Y12 inhibitors has been shown to have shifted from clopidogrel and prasugrel towards ticagrelor from 2015 to 2017. The decrease in clopidogrel prescription and the parallel increased prescription of novel P2Y12 inhibitors has already been demonstrated in Austria between 2009 and 2014 and reflects implementation of ACS treatment guidelines [2]. However, clopidogrel remained the single most common P2Y12 inhibitor employed for ACS, especially in patients aged ≥70 years.

Comparing our results with findings from the epidemiological study conducted between 2009 and 2014 in Austria, the approximately 50% drug discontinuation of P2Y12 therapy after 12 months is similar. This is primarily caused by the limited period of reimbursement of prasugrel and ticagrelor in Austria. The observed prolonged prescription of clopidogrel is consistent with previously reported data in Austria. Of note, the prescription charges per package of clopidogrel, prasugrel, and ticagrelor are not different in Austria. Thus, the selection of P2Y12 inhibitor treatment is attributed primarily to the center where these acute patients have been admitted to and regional differences that exist. Furthermore, P2Y12 inhibitor cessation after six months occurs in approximately 20% of patients treated with clopidogrel. Due to lack of information on anatomical, procedural, and clinical conditions or therapy management strategies, this has to remain speculative. In this context, it is important to note that no data for aspirin prescriptions are available because costs are below the threshold of co-pay by the health care system.

Previous reports have shown a gender difference in treatment choice of P2Y12 inhibitors [20,21]. This was also demonstrated in the present study, where less than 28% of patients receiving ticagrelor and less than 20% of those receiving prasugrel were female. In comparison, the percentage of female patients receiving clopidogrel was almost 39%. Of note, the outcome after ACS was worse in women and the risk of an endpoint in men was significantly lower. In this respect, our second new finding is that women with ACS in the youngest quartile group had a higher hazard for ACS or death compared to men at the same age. It is unclear if this sex-related difference in overall clinical outcome is directly related to the preferences in the prescription of DAPT.

The number of prescriptions of CV, anti-diabetic, and anti-obstructive medicines was consistent with our previous findings, except for HMG CoA reductase inhibitors. In the present study, almost 50% of ACS patients were under statin treatment, which is twice as much as before. Nonetheless utilization remained lower than recommended in treatment guidelines [22,23]. Statin intolerance and clinical contraindications may explain the underuse of statin in ACS patients; however, the main reasons of underutilization and non-adherence to guidelines need further examination.

In contrast to our findings, the RENAMI subgroup analysis of patients with diabetes reported a reduced all-cause mortality and bleeding in subjects when treated with ticagrelor compared to those treated with prasugrel [24].

There are several limitations in the present study. To identify patients, ICD-10 coding was used; however, we were not able to differentiate between patients with STEMI and NSTEMI. Another limitation was the retrospective design. Additionally, there is a lack of data on clinical symptoms or diagnostic tests, e.g., PCI or echocardiography, which are not included in the claims database of the health insurance funds. In addition, our data do not provide information on adverse drug events, such as bleeding and stent thrombosis. Furthermore, no data on P2Y12 inhibitor intake at the time of an event, drug discontinuation, or patient’s adherence were available.

## 5. Conclusions

In conclusion, our analysis suggests that the choice of P2Y12 inhibitor therapy impacts the risk of ACS recurrence or death. Clopidogrel is the most prescribed P2Y12 inhibitor, but an increased use of ticagrelor is evident. The drug utilization with P2Y12 inhibitors after ACS is consistent with guideline recommendations in Austria.

## Figures and Tables

**Figure 1 jcm-09-03398-f001:**
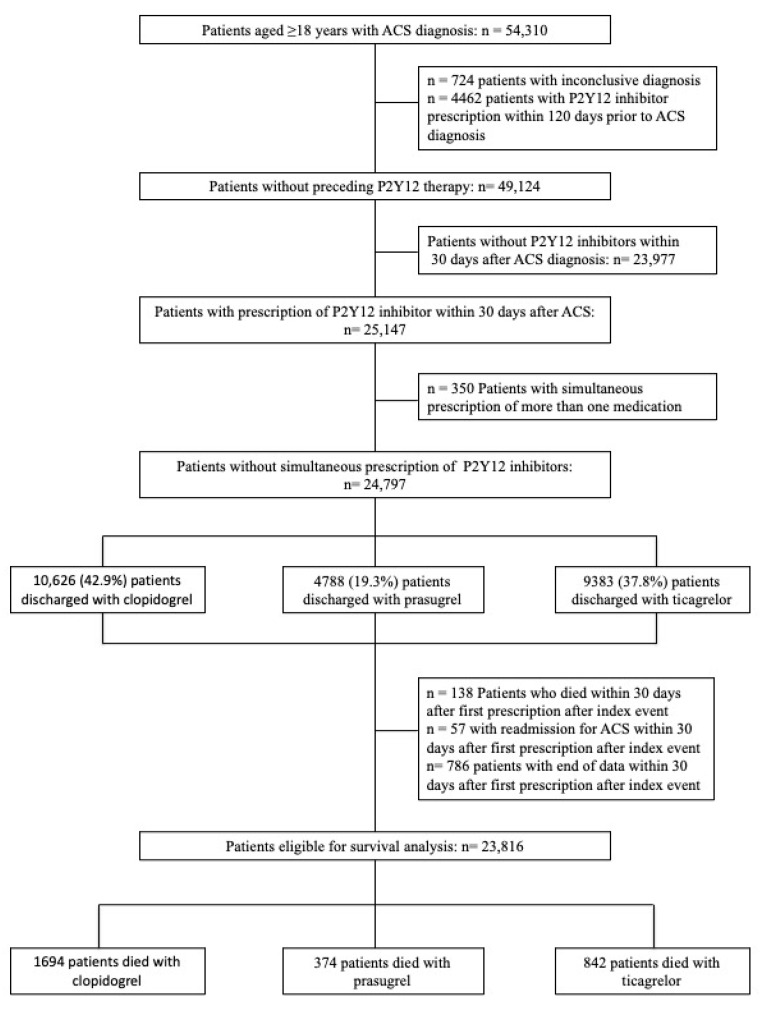
Flowchart of patients with discharge diagnosis of acute coronary syndrome.

**Figure 2 jcm-09-03398-f002:**
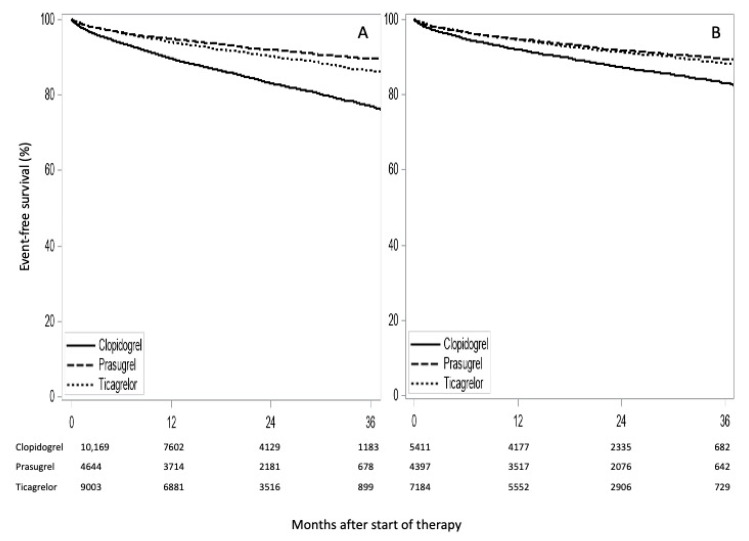
Event-free survival (recurrence of ACS or death) for all patients (**A**) and for patients aged ≤75 years (**B**) treated with clopidogrel, prasugrel, or ticagrelor. Numbers at risk are indicated. (A) *p* < 0.001 (log-rank test); (B) *p* < 0.001 (log-rank test).

**Figure 3 jcm-09-03398-f003:**
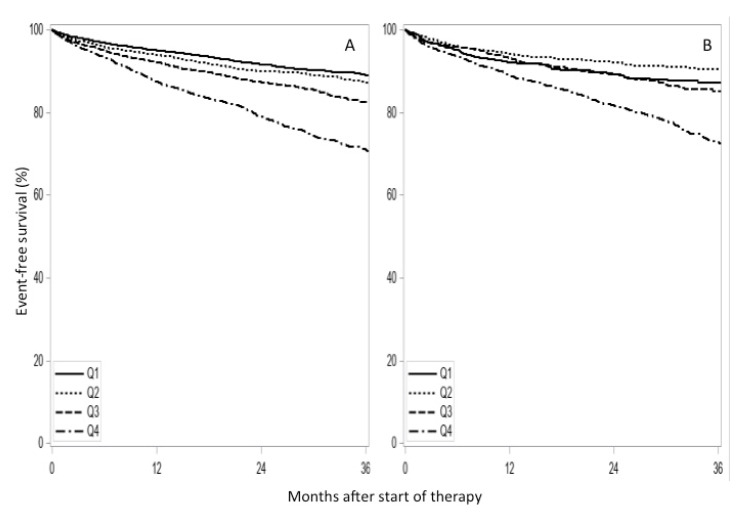
Event-free survival (recurrence of ACS or death) for male (**A**) and female (**B**) patients treated with clopidogrel, prasugrel, or ticagrelor. Numbers at risk are indicated. Q1, Q2, Q3, and Q4 quartile groups comprise patients aged 18–55, 56–66, 67–76, and ≥77 years, respectively. *p* < 0.001 (log-rank test).

**Table 1 jcm-09-03398-t001:** Patients characteristics and disease-specific medication before index ACS admission.

Characteristics	Clopidogrel (*n* = 10626)	Prasugrel (*n* = 4788)	*p*-Value Clopi vs. Prasu	Ticagrelor (*n* = 9383)	*p*-Value Clopi vs. Tica
Age—median years (IQR)	74 (64; 81)	57 (51; 65)	*p* < 0.001	63 (54; 73)	*p* < 0.001
Female sex—*n* (%)	4103 (38.6)	933 (19.5)	*p* < 0.001	2622 (27.9)	*p* < 0.001
Cardiovascular medication—*n* (%)	5585 (52.6)	1481 (30.9)	*p* < 0.001	3880 (41.4)	*p* < 0.001
HMG CoA reductase inhibitors—*n* (%)	5926 (55.8)	1961 (41.0)	*p* < 0.001	4579 (48.8)	*p* < 0.001
Drugs for obstructive airway diseases—*n* (%)	1026(9.7)	247 (5.2)	*p* < 0.001	616(6.6)	*p* < 0.001
Anti-diabetic medicines—*n* (%)	1743 (16.4)	506 (10.6)	*p* < 0.001	1278 (13.6)	*p* < 0.001
NOAC—(%)	1221 (11.5)	9 (0.2)	*p* < 0.001	34 (0.4)	*p* < 0.001

ACS, acute coronary syndrome, IQR, interquartile ranges; *n*, number of patients; NOAC, non-vitamin-K oral anticoagulant.

**Table 2 jcm-09-03398-t002:** Prescription rate (%) of P2Y12 inhibitors according to age, calendar year, and drug survival over 24 months.

	Clopidogrel	Prasugrel	*p*-Value Clopi vs. Prasu	Ticagrelor	*p*-Value Clopi vs. Tica
**Age**					
Age 18–39 (*n* = 454)	14.8%	38.8%	*p* < 0.001	46.5%	*p* < 0.001
Age 40–49 (*n* = 2374)	16.2%	37.4%	*p* < 0.001	46.5%	*p* < 0.001
Age 50–59 (*n* = 5749)	23.6%	31.8%	*p* < 0.001	44.6%	*p* < 0.001
Age 60–69 (*n* = 5963)	37.8%	20.9%	*p* < 0.001	41.3%	*p* < 0.001
Age 70–79 (*n* = 6205)	55.5%	9.6%	*p* < 0.001	34.9%	*p* < 0.001
Age 80–89 (*n* = 3532)	75.3%	1.5%	*p* < 0.001	23.2%	*p* < 0.001
Age ≥ 90 (*n* = 520)	88.5%	0.2%	*p* < 0.001	11.3%	*p* < 0.001
**Calendar year**					
2015 (*n* = 8040)	44.3%	21.9%	*p* < 0.001	33.7%	*p* < 0.001
2016 (*n* = 8417)	43.5%	18.7%	*p* < 0.001	37.8%	*p* < 0.001
2017 (*n* = 8340)	40.8%	17.4%	*p* < 0.001	41.9%	*p* < 0.001
**Drug survival**					
6 months	77.2%	95.0%	*p* < 0.001	91.7%	*p* < 0.001
12 months	47.1%	52.9%	*p* < 0.001	47.7%	*p* < 0.001
18 months	23.5%	11.4%	*p* < 0.001	8.4%	*p* < 0.001
24 months	17.3%	6.8%	*p* < 0.001	5.1%	*p* < 0.001

*n*, number of patients.

**Table 3 jcm-09-03398-t003:** Predictors of recurrence of ACS or death.

(*n* = 23816)	P2Y12 Inhibitors	Crude HR (95% CI)	Adjusted HR (95%CI)
Medication	prasugrel	0.49 (0.44; 0.55)	0.70 (0.61; 0.79)
	ticagrelor	0.57 (0.52; 0.62)	0.70 (0.64; 0.77)
	clopidogrel	1	1
Age (median 61 years)	56–66 (2.Qu)	1.01 (0.89; 1.16)	
(median 72 years)	67–76 (3.Qu)	1.44 (1.28; 1.60)	
(median 82 years)	≥77 (4.Qu)	2.18 (1.97; 2.41)	
(median 51 years)	18–55 (1.Qu)	1	
Age, males			
(median 61 years)	56–66 (2.Qu)		1.02 (0.88; 1.18)
(median 72 years)	67–76 (3.Qu)		1.29 (1.12; 1.47)
(median 82 years)	≥77 (4.Qu)		1.93 (1.69; 2.20)
(median 51 years)	18–55 (1.Qu)		1
Age, females			
(median 61 years)	56–66 (2.Qu)		0.63 (0.48; 0.84)
(median 72 years)	67–76 (3.Qu)		0.93 (0.74; 1.17)
(median 82 years)	≥77 (4.Qu)		1.31 (1.07; 1.61)
(median 51 years)	18–55 (1.Qu)		1
Sex	male vs. female	0.87 (0.80; 0.94)	
Sex, 18–55 years	male vs. female		0.81 (0.67; 1.00)
Sex, 56–66 years	male vs. female		1.31 (1.02; 1.69)
Sex, 67–76 years	male vs. female		1.13 (0.93; 1.37)
Sex, ≥77 years	male vs. female		1.20 (1.10; 1.34)
Cardiovascular medicines	yes vs. no	1.41 (1.32; 1.52)	1.18 (1.08; 1.29)
HMG CoA reductase inhibitors	yes vs. no	1.20 (1.12; 1.29)	0.98 (0.90; 1.07)
Drugs for obstructive airway diseases	yes vs. no	1.57 (1.39; 1.77)	1.36 (1.21; 1.54)
Anti-diabetic medicines	yes vs. no	1.46 (1.33; 1.60)	1.33 (1.21; 1.47)
Non-Vitamin K oral anticoagulant	yes vs. no	1.40 (1.21; 1.65)	0.93 (0.79; 1.09)
Calendar year	2016 vs. 2015	1.01 (0.92; 1.01)	1.01 (0.92; 1.10)
	2017 vs. 2015	1.14 (1.03; 1.27)	1.17 (1.05; 1.30)

HR, hazard ratio; CI, confidence interval; ref, reference value; Qu, quartile group.

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
