# Peer review of "Use of Clopidogrel, Prasugrel, or Ticagrelor and Patient Outcome after Acute Coronary Syndrome in Austria from 2015 to 2017"

_jcm, 2020, doi:10.3390/jcm9113398_

Round 1
Reviewer 1 Report
Safoura Sheikh Rezaei et al present an observational study on interaction between P2Y12 inhibitors and outcomes after an ACS events. The population presented is quite big, but comes only from a single nation (Austria) and completeness of data is quite low.
They conclude suggesting a protective association between prasugrel and CV outcome at unadjusted analysis that tend to disappear after multivariate analysis. However, to be more clear, an analysis directly comparing ticagrelor and prasugrel cohort should be performed and presented (for example excluding patients treated with clopidogrel and comparing ticagrelor vs prasugrel subgroup in cox survival regression analysis). Furthermore, the authors underline interactions between female sex, a lower prescription rate of Tica and prasugrel vs clopidogrel and a worse clinical outcome.
Even if the number of patients involved in the present analysis is huge, many significant limitations have to be underlined:
> the primary endpoint seems quite inappropriate for this comparison. The role of bleeding should be considered at least as secondary endpoint and absence of data on this complication clearly limits any interpretation of results on DAPT therapy. An other important point is absence on data on concomitant need for anticoagulation that may contribute to bleeding and death.
> The number of patients considered in the present analysis when compared to those initially selected from the database is very low. This may lead to significant selection biases. Authors should explain why they excluded from the analysis patients experiencing the outcome (death or ACS) within the first 30 days from the index events (see figure 1). It is well known that bleeding events often occur in the first month. Furthermore, a significant percentage of re-AMI or stent thrombosis may occurr in the first months after the index events.
>The interaction between diabetic status (as inferred by prescribed therapy) and outcome according to P2Y12 type should be analysed. Their results should be compared to the RENAMI subgroup analysis (with propensity match) (see: Eur Heart J Acute Cardiovasc Care. 2019 Sep;8(6):536-542. doi: 10.1177/2048872618802783. Epub 2018 Oct 1)
>No data on switch between the different p2y12 inhibitors after discharge has been presented. It may plays a significant role in outcome interpretation. Furthermore drug-survival analysis is clearly influenced by ACS re-occurence during follow-up apart from long term dapt.
> conclusion on prasugrel vs ticagrelor outcome are quite weak considering the limitation in available data from the dataset (absence of data on type of ACS, type coronary artery disease, anticoagulation therapy, discharge ejection fraction) and consequent covariates considered in the multivariate analysis. For example it can be speculated that a higher number of patients in the prasugrel group were admitted for STEMI considering limitation on prescription derived from the accost study.
> finally, the cohort refers to a period between 2015 and 2017. Recent observational and randomized data may influence p2y12 prescriptions in these patients within last years.
Author Response
Safoura Klopprogge (Sheikh Rezaei), MD
Michael Wolzt, MD
Medical University of Vienna
Department of Clinical Pharmacology
Währinger Gürtel 18-20
1090 Vienna, Austria
__________________________________________
Vienna, 1st October, 2020
Revision of the manuscript Manuscript ID: jcm-951427 (“Use of clopidogrel, prasugrel, or ticagrelor and patients outcome after an acute coronary syndrome in Austria from 2015 to 2017.”).
Dear Ms. Assistant Editor
We are grateful for the opportunity to submit a revision of our manuscript titled “Use of clopidogrel, prasugrel, or ticagrelor and patients outcome after an acute coronary syndrome in Austria from 2015 to 2017.”, manuscript number JCM-951427.
We would like to thank you and the reviewer for the efforts and valuable comments on our manuscript. These helped very much to improve the revised version. Please see the detailed point-by-point response below.
We would be delighted if the revised manuscript is acceptable for publication in your distinguished journal in the present form.
Yours sincerely,
Michael Wolzt
Safoura Klopprogge (Sheikh Rezaei)
Point-by-Point Response for JCM -951427
Use of clopidogrel, prasugrel, or ticagrelor and patients outcome after an acute coronary syndrome in Austria from 2015 to 2017.
________________________
Reviewer 1:
Safoura Sheikh Rezaei et al present an observational study on interaction between P2Y12 inhibitors and outcomes after an ACS events. The population presented is quite big, but comes only from a single nation (Austria) and completeness of data is quite low.
They conclude suggesting a protective association between prasugrel and CV outcome at unadjusted analysis that tend to disappear after multivariate analysis. However, to be more clear, an analysis directly comparing ticagrelor and prasugrel cohort should be performed and presented (for example excluding patients treated with clopidogrel and comparing ticagrelor vs prasugrel subgroup in cox survival regression analysis). Furthermore, the authors underline interactions between female sex, a lower prescription rate of Tica and prasugrel vs clopidogrel and a worse clinical outcome.
Thank you for this comment. As suggested by the reviewer we compared clopidogrel vs ticagrelor and prasugrel vs ticagrelor in a re-parameterized multivariable regression model.
“A re-parameterization of the adjusted model allows to compare clopidogrel and prasugrel with ticagrelor and results in a HR 1.76 (95% CI 1.62;1.91) for clopidogrel vs ticagrelor and 0.86 (95% CI 0.76;0.97) for prasugrel vs. ticagrelor.”
Even if the number of patients involved in the present analysis is huge, many significant limitations have to be underlined:
> the primary endpoint seems quite inappropriate for this comparison. The role of bleeding should be considered at least as secondary endpoint and absence of data on this complication clearly limits any interpretation of results on DAPT therapy. Another important point is absence on data on concomitant need for anticoagulation that may contribute to bleeding and death.
We agree with the reviewer that the role of bleeding as secondary endpoint would be valuable, however no information is available for bleeding in the database of the health insurance funds and the national death registry maintained by Statistics Austria. We have stated this limitation in our Limitation section.
“In addition, our data does not provide information on adverse drug events such as bleeding, stent thrombosis, on real intake of the medications or patient’s adherence.”
> The number of patients considered in the present analysis when compared to those initially selected from the database is very low. This may lead to significant selection biases. Authors should explain why they excluded from the analysis patients experiencing the outcome (death or ACS) within the first 30 days from the index events (see figure 1).
We agree with the reviewer that more information about the excluded 195 patients would be valuable. We therefore added the following sentence to the Results section of our paper.
“We excluded 195 patients who died or had a re-ACS within 30 days after first prescription of P2Y12 inhibitors after index-ACS, since reimbursement data are capped on a quarterly basis. Also, in some cases patients are discharged with promotional offer of one month's free medication and therefore we used the time frame of 30 days “
It is well known that bleeding events often occur in the first month. Furthermore, a significant percentage of re-AMI or stent thrombosis may occur in the first months after the index events.
As mentioned above the lack of information on bleeding, re-AMI or stent thrombosis is a limitation of the manuscript, which is discussed in Limitation section.
“In addition, our data does not provide information on adverse drug events such as bleeding, stent thrombosis, on real intake of the medications or patient’s adherence.”
>The interaction between diabetic status (as inferred by prescribed therapy) and outcome according to P2Y12 type should be analysed. Their results should be compared to the RENAMI subgroup analysis (with propensity match) (see: Eur Heart J Acute Cardiovasc Care. 2019 Sep;8(6):536-542. doi: 10.1177/2048872618802783. Epub 2018 Oct 1)
As suggested, we calculated the interaction of anti-diabetic medicine with P2Y12 inhibitors and discussed the RENAMI subgroup analysis in discussion.
“The interaction of anti-diabetic medicines and P2Y12 inhibitor medication was not added to the model in Table 3, since it was not statistically significant (p=0.260).”
“In contrast to our findings, the RENAMI subgroup analysis of patients with diabetes reported a reduced all-cause mortality and bleeding in subjects when treated with ticagrelor compared to those with prasugrel.[24]”
>No data on switch between the different p2y12 inhibitors after discharge has been presented. It may plays a significant role in outcome interpretation. Furthermore drug-survival analysis is clearly influenced by ACS re-occurence during follow-up apart from long term dapt.
Thank you for this valuable comment. In the present study, 350 patients were detected with simultaneous prescription of more than one medication (Table 1). The detailed switch between P2Y12 inhibitors was added as suggested.
“350 patients with simultaneous prescription of P2Y12 inhibitors were excluded. From these, 250 patients switched between clopidogrel and ticagrelor, 71 patients between clopidogrel and prasugrel, 57 patients between ticagrelor and prasugrel, and 7 patients switched between clopidogrel, ticagrelor, and prasugrel.”
> conclusion on prasugrel vs ticagrelor outcome are quite weak considering the limitation in available data from the dataset (absence of data on type of ACS, type coronary artery disease, anticoagulation therapy, discharge ejection fraction) and consequent covariates considered in the multivariate analysis. For example it can be speculated that a higher number of patients in the prasugrel group were admitted for STEMI considering limitation on prescription derived from the accost study.
We agree with the reviewer that our study has several limitations. As already mentioned above, these has been discussed in Limitation section.
> finally, the cohort refers to a period between 2015 and 2017. Recent observational and randomized data may influence p2y12 prescriptions in these patients within last years.
Thank you for this comment. We only had access to data from 2015 to 2017 from database of the health insurance funds and the national death registry maintained by Statistics Austria.
Reviewer 2 Report
Herein, Rezaei et al report an interesting analysis of the long-term impact of P2Y12 inhibitor therapy in patients with acute coronary syndrome. This data from the Austrian Health Insurance records show most patients (43%) were on prescribed clopidogrel. Compared to clopidogrel, use of prasugrel or ticagrelor was associated with reduced recurrent coronary events. The findings are very timely and would be well received. However, there are some issues that the authors would need to address.
Major comments
- The ethics statements lack information on whether the patients consented to being part of this study. I think the authors should make this clear.
- The authors have not indicated whether there was any statistical significant difference between the baseline characteristics, shown in Table 1, among the groups. I think this should be improved. There should be an extra column to indicate the level of significance. Similarly, Table 2 should be improved. I recommend clarifying whether the reduction in frequency of drug use changed over time among the three groups.
- The reporting of drug survival (section 3.1) is misleading. The authors should present and compare the median change in survival among the groups, rather than between consecutive timepoints within the same therapy drug.
- In the regression models, clopidogrel was used as the reference therapy. However, there is no rationale provided for doing so.
Minor comments
- The authors note that metric variables are described by medians and interquartile ranges. However, they have not indicated whether these data were skewed and if they performed any normality testing.
- Line 105, the authors should write 49124 in full.
- In Figure 2 and 3, the authors should provide the relevant p-values.
- The retrospective nature of the present study should be noted in the limitations
Author Response
Reviewer 2:
Herein, Rezaei et al report an interesting analysis of the long-term impact of P2Y12 inhibitor therapy in patients with acute coronary syndrome. This data from the Austrian Health Insurance records show most patients (43%) were on prescribed clopidogrel. Compared to clopidogrel, use of prasugrel or ticagrelor was associated with reduced recurrent coronary events. The findings are very timely and would be well received. However, there are some issues that the authors would need to address.
Major comments
- The ethics statements lack information on whether the patients consented to being part of this study. I think the authors should make this clear.
Thank you for this comment.
This study was approved by the Ethics Committee of the Medical University of Vienna (EK-No. 2042/2018) and performed according to the Declaration of Helsinki. Patients informed consent was not required due to the retrospective design of the present study. The present investigation was performed in cooperation with the Pharmacoeconomics Advisory Council of the Austrian Sickness Fund.
- The authors have not indicated whether there was any statistical significant difference between the baseline characteristics, shown in Table 1, among the groups. I think this should be improved. There should be an extra column to indicate the level of significance. Similarly, Table 2 should be improved. I recommend clarifying whether the reduction in frequency of drug use changed over time among the three groups.
Due to the high sample size we originally refrained from presenting p-values, since the tests were expected to be over-powered. In addition, hypothesis testing is not really an issue for answering our research qeustions, but rather a concise description of the observed effects. Therefore, in the original manuscript we presented confidence intervals for all estimated effect measures in order to quantify their uncertainty.
As suggested by the reviewer we calculated the demanded p-values and, expectedly, they are all below 0.001. However, we do not think that these add any information to the results originally presented, e.g. confidence intervals, and therefore would prefer not to add them. We leave it to the Assistant Editor to decide whether these p-values should be presented or not.
- The reporting of drug survival (section 3.1) is misleading. The authors should present and compare the median change in survival among the groups, rather than between consecutive timepoints within the same therapy drug.
Done as suggested.
“Drug survival is shown in Table 2. During the observation of 24 months the median drug survival for clopidogrel, prasugrel, and ticagrelor was 11.8 (IQR 6.4-17.0), 12.1 (IQR 11.3-13.5), and 12.0 (IQR 11.1-12.9) months, respectively.”
- In the regression models, clopidogrel was used as the reference therapy. However, there is no rationale provided for doing so.
Clopidogrel has been used as control treatment in randomized trials of P2Y12 inhibitors and is considered as standard medical therapy in patients with ACS, therefore we used clopidogrel as reference therapy in our study. As suggested by reviewer one and two, we compared clopidogrel vs ticagrelor and prasugrel vs ticagrelor in a re-parameterized multivariable regression model.
“A re-parameterization of the adjusted model allows to compare clopidogrel and prasugrel with ticagrelor and results in a HR 1.76 (95% CI 1.62;1.91) for clopidogrel vs ticagrelor and 0.86 (95% CI 0.76;0.97) for prasugrel vs. ticagrelor.”
Minor comments
- The authors note that metric variables are described by medians and interquartile ranges. However, they have not indicated whether these data were skewed and if they performed any normality testing.
Thank you for this comment. In large samples as ours, pre-tests for normal distribution are known to detect deviations from normal distribution that are irrelevant, i.e. that would not necessitate the use of non-parametric methods. Therefore we checked for normal distribution using graphical methods (Boxplots, QQ-plots). The first sentence in the statistical analysis section was amended accordingly.
“Metric variables are reported by medians and interquartile ranges (IQR) due to non-normal distribution (graphically checked) and compared between P2Y12 inhibitors using a Kruskal-Wallis test. Categorical variables are described by absolute and relative frequencies and compared between P2Y12 inhibitors using Chi-square tests.“
- Line 105, the authors should write 49124 in full.
Done as suggested.
“Forty-nine thousand one hundred twenty-four P2Y12 inhibitor-naive patients with ACS were identified (Figure 1).”
- In Figure 2 and 3, the authors should provide the relevant p-values.
p-values was added to the legend of Figure 2 and 3 as suggested. The statistical section was also added accordingly.
“The distribution of time to event, which was defined as re-admission for ACS or death, was estimated using the product-limit method, censoring for end of data availability, and compared between P2Y12 inhibitors using the log-rank test.“
“Figure 2A: p<0.001 (log-rank test), Figure 2B: p<0.001 (log-rank test).”
„Figure 3A: p<0.001 (log-rank test), Figure 3B: p<0.001 (log-rank test).“
- The retrospective nature of the present study should be noted in the limitations
“There are several limitations in the present study. To identify patients ICD-10 coding was used, however, we were not able to differentiate between patients with STEMI and NSTEMI. Another limitation was the retrospective nature of the study. Also, there is a lack of data on clinical symptoms or diagnostic tests, e.g. PCI or echocardiography, which are not included in the claims database of the health insurance funds. In addition, our data does not provide information on adverse drug events such as bleeding, on real intake of the medications or patient’s adherence.”
Reviewer 3 Report
This a well written manuscript of the P2Y12 inhibitor use in Austria. Still there is a lot of major limitations in this study. The conclusions can not be based on this kind of data. Data is based only to ICD codes and medication. Sex, age and previous medication is not data enough to conclude which medication performs better or best.
Limitations:
- ICD codes I24.9, I25.0, 25.1 and I20.8x are not ACS codes. Patients should be diveded according codes at least to chronic, NSTEMI and STEMI.
- How the follow-up was done. It is not described.
- Risk profile of patients should be known (anemia, anticoagulation, gerastenia, bleeding risk, diabetes, kidney disease, liver disease…) to make any study between different medication. After this you can match populations. Then it is possible to compare different medications.
- The number of revascularisation in different medication and groups is essential information. Must be included.
In conclusion there is too many limitations to make this kind of conclusion that is made in this manuscript. The only conclusion that can be made is that the use of ticagrelor is higher 2017 than 2015. In patients that prasugrel and ticagrelor is used the morality/reischemia is lower. Conclusion should be correted or matching for risks and revascularisation should be performed .
Author Response
Reviewer 3:
This a well written manuscript of the P2Y12 inhibitor use in Austria. Still there is a lot of major limitations in this study. The conclusions can not be based on this kind of data. Data is based only to ICD codes and medication. Sex, age and previous medication is not data enough to conclude which medication performs better or best.
Limitations:
- ICD codes I24.9, I25.0, 25.1 and I20.8x are not ACS codes. Patients should be diveded according codes at least to chronic, NSTEMI and STEMI.
Since data were collected from 2015 to 2017 the ICD discharge diagnoses were according to ECS Guidelines from 2017 and before. Also, in the database of the health insurance funds and the national death registry maintained by Statistics Austria no information on STEMI or NSTEMI is available. We have stated this limitation in our Limitation section.
- How the follow-up was done. It is not described.
Thank you for this valuable comment. We added this information accordingly in statistical methods section.
- Risk profile of patients should be known (anemia, anticoagulation, gerastenia, bleeding risk, diabetes, kidney disease, liver disease…) to make any study between different medication. After this you can match populations. Then it is possible to compare different medications.
We agree with the reviewer that more information about patients risk profile would be valuable. We have stated this limitation in our Limitation section.
- The number of revascularisation in different medication and groups is essential information. Must be included.
In conclusion there is too many limitations to make this kind of conclusion that is made in this manuscript. The only conclusion that can be made is that the use of ticagrelor is higher 2017 than 2015. In patients that prasugrel and ticagrelor is used the morality/reischemia is lower. Conclusion should be correted or matching for risks and revascularisation should be performed .
Again, this information was not available and is discussed in the Limitation section.
Round 2
Reviewer 2 Report
The authors have addressed all my concerns. However, on the ethical issues, can the authors clarify whether patient identities were concealed?
Author Response
Thank you.
Reviewer 3 Report
This manuscript is better. This study has limitations and due the limitations it can not be better. It is now written in this manuscript. I still think that differences between medication can be explained with different kind of patients. The difference between medications can be studied mainly on randomised studies or like in "change dapt" study, where protocol is totally different in time periods. This should be recognised in this manuscript. Now it is possible that medications are used in totally different patients populations eg clopidogrel used in patients with bleeding risk etc.
Author Response
Thank you. We have added the limitations in discussions.